# Oral magnesium supplementation for leg cramps in pregnancy—An observational controlled trial

Carla Adriane Leal de Araújo[1], Suélem Barros de Lorena[2‡], Guilherme Camelo de Sousa Cavalcanti[2‡], Gabriel Landim de Souza Leão[2‡], Geraldo Padilha Tenório[2‡], João Guilherme B. Alves[1]*

1 Department of Pediatrics, Instituto de Medicina Integral Prof. Fernando Figueira (IMIP), Recife, Pernambuco, Brazil, 2 Department of Pediatrics, Faculdade Pernambucana de Saúde (FPS), Recife, Pernambuco, Brazil

☯ These authors contributed equally to this work.
‡ These authors also contributed equally to this work.
* joaoguilherme@imip.org.br

**Data Availability Statement:** All relevant data are within the mansucript and its Supporting Information files.

## Abstract

### Background

Oral magnesium for leg cramps treatment in pregnancy is a controversial issue according to recent Cochrane systematic review. This study aims to evaluate the efficacy of $Mg^{++}$ supplementation in leg cramps treatment in pregnancy.

### Methods

This observational clinical trial studied 132 pregnant women with leg cramps in the first trimester of pregnancy. At baseline, 74 (56.3%) had two leg cramps episodes per week, 28 (21.1%) three episodes, 13 (9.8%) four episodes and 9 (6.8%) five or more episodes. They were randomized 1:1 to 300 mg/day of oral $Mg^{++}$ citrate (n = 66) or placebo (n = 66). The primary outcome was the frequency of leg cramps episodes per week reported by pregnant women. Secondary outcomes were the ocurrence of leg cramps and oral magnesium side effects.

### Results

130 pregnant women completed the study and the two groups were comparable according to some sociodemographic and clinical characteristics. After 4 weeks of intervention it was observed a 28.4% (39/132) (CI 95%: 20.9–37.0) reduction of leg cramps in all participants and no difference between the two groups was found; reduction of 27.2% (18/66) (CI 95%: 17.0–39.6) in Mg++ group and 32.8% (21/66) (CI 95%: 21.6–45.7) in the placebo group. The OR of leg cramps was 1.3 (CI 95%: 0.6–2.9), p = 0.527, taking the placebo group as reference. Among pregnant women who remained with leg cramps the mean of leg cramps episodes per week showed no significance difference between the Mg++ and placebo groups; t-student test: p = 0.408. Four pregnant women showed gastrointestinal side effects; 2 in each group had nauseas and diarrhoea.

**Funding:** Funded by the Bill & Melinda Gates Foundation (OPP1107597) and Conselho Nacional de Desenvolvimento Científico e Tecnólogico, CNPq (401609-2013-8).

**Competing interests:** The authors have declared that no competing interests exist.

## Conclusion

Oral magnesium supplementation during pregnancy did not reduce the ocurrence and frequency of episodes of leg cramps.

## Introduction

Leg cramps are involuntary painful skeletal muscle contractions lasting from seconds to minutes, often nocturnal, and frequently involve the gastrocnemius [1,2]. About 30–50% of pregnant women experience leg cramps at least twice a week during the third trimester [2,3]. The etiology of cramps during pregnancy is unclear, but is believed to be due to overload of the ankle plantar flexors, excessive exercise, metabolic disorders, circulatory problems, underlying medical conditions, nutritional deficiencies (vitamins E and D) or electrolyte imbalances (eg. magnesium, calcium and sodium) [1,2,4].

Many therapies for leg cramps have been tried as gabapentin, pycnogenol, electrolytes and vitamins (magnesium, calcium, sodium, and vitamin E and vitamin D), massage, muscle stretching, relaxation, heat therapy and dorsiflexion of the foot [2]. However, there are still no consistent conclusions for treating leg cramps in pregnancy. [2]. Magnesium plays an important role in many metabolic reactions, neuronal excitability and muscle function [5]. Magnesium deficiency enhances neuromuscular transmission. Magnesium therapy has been shown to be effective in eclampsia-related seizures [6]. For this reason some studies have suggested a beneficial role of magnesium in leg cramps. Even more because magnesium requirements increase during pregnancy [7]. Magnesium supplementation has been indicated as a therapy capable of minimizing fetal growth restrictions, reducing the risk of preeclampsia and favoring newborn weight gain, although there is no strong evidence of these benefits [8]. Besides insufficient magnesium intake is common, especially in low-income regions [9].

Some systematic reviews showed that is unclear whether oral magnesium provides an effective treatment for leg cramps and large well-conducted randomised controlled trials are needed to answer the question of leg cramps treatment [2, 10, 11, 12]. This is even more important for low-income regions where insufficient magnesium intake is common [9]. This study aims to evaluate the efficacy of oral magnesium supplementation in the treatment of cramps during pregnancy in a low-income region.

## Methods

### Study design

This observational blinded controlled trial investigated the effect of oral magnesium citrate supplementation for leg cramps in pregnant women. This trial tests the hypothesis that oral magnesium supplementaton during pregnancy may reduce the frequency od leg cramps episodes. The intervention lasted 4 weeks and was completed between November 2015 and January 2018. This study was part of the Brazil MAGnesium trial [13] and registered at ClinicalTrials.gov (Identifier NCT02032186).

### Setting and participants

The study took place at Instituto de Medicina Integral Prof. Fernando Figueira (IMIP), Brazil. IMIP registers about 6,000 deliveries per year. Pregnant women who attended the antenatal care clinic at the Department of Obstetrics, IMIP, were invited to join this study. Inclusion

criteria were pregnant women aging between 18–45 years, gestational age between 12 and 20 weeks, a single gestation and currently residents of the city of Recife. Gestation age was based on the last menstrual period among women with a regular menstrual cycle or by first-trimester pregnancy dating ultrasound. Exclusion criteria were uncontrolled known hyperthyroidism, any type of known active parathyroid disease, chronic diarrheal disease, chronic kidney disease, defined by an estimated glomerular filtration rate below 60 mL / min / 1.73 $m^2$, as determined by the initial assessment or by known history, Mg ++ serum concentration at baseline > 2.6 mg / dL.

## Randomization and intervention

Randomization was performed in a 1:1 ratio using a table of random numbers, prepared by a researcher who did not participate in the data collection. These numbers were generated in a computer by Random Allocation Software 2.0 program. Allocation concealment was ensured, as the referred researcher did not release the randomization code until the participants were recruited into the trial after all baseline measurements were completed.

Consenting pregnant women received a 4 weeks magnesium citrate capsule (300 mg elemental magnesium citrate per capsule) or a daily placebo capsule identical in colour and shape. Both capsules were manufactured by IMIP's Department of Pharmacology. The study medication packages were supplied with sequential numbers. Code break envelopes were supplied to the lead pharmacist but not available for the investigation team. Each pack was individually prescribed for each participant. Compliance, adverse events, and clinical intercurrences were monitored by the research team at routine prenatal visit during the intervention. Adherence to treatment was defined as the ingestion of at least 80% of the prescribed dose for 30 days.

The criteria for discontinuation of the study were: symptoms reported by the patient or clinical signals due to the intake of the magnesium capsules or the cancellation of prenatal care at IMIP.

## Outcomes

The primary outcome was the frequency of leg cramps which was defined as painful, involuntary contraction of muscles occurring at rest, mostly at night, and causing a palpable knot in the muscle, recorded at least twice a week. The frequency was measured as the number of leg cramps per week. Secondary outcomes were the presence of leg cramps episodes and oral magnesium side effects. All this informations were taken from the research diaries previously given to participants.

## Ethical considerations

All participants were informed about the data confidentiality, and were informed about their capacity to withdraw from the study. All pregnant women provided written informed consent. The study was approved by the IMIP's Committee on Research (document number 4033), and was registered in the ClinicaTrials.gov (NCT 02032186).

## Data analysis

The sample size calculation was calculated based on Dahle et al (12) and Supakatisant C & Phupong (13) trials to compare 50% reduction of leg cramps between intervention and placebo groups. It was assumed a 5% bilateral alpha error, a 80% power and an expected 35% percentage of cramps in the intervention group. With adjustments for a withdrawal rate of 10%, a

minimum of 66 women in each group were required. Stata version 12.1 was used for statistical analysis. Chi-squared test for categorical variables and independent *t*-test for continuous variables were used when appropriate. It was considered a p < 0.05. Intent-to-treat analysis was performed.

## Results

A total of 394 pregnant women were screened and 132 enrolled according to the inclusion criteria. At baseline, 74 (56.0%) had two leg cramps episodes per week, 28 (21.2%) three episodes, 7 (9.8%) four episodes and 23 (13.0%) five or more episodes; 66 pregnant women were assigned to the Mg++ group (300 mg per day) and 66 were assigned to the placebo (Fig 1). Two pregnant women of placebo group were lost to follow-up but 132 participants were included in the intention-to-treat analysis.

The groups showed no significant diferences with respect to age, years of study, employment, income, parity, body mass index, gestational age, number of leg cramps episodes per week and serum magnesium level (Table 1).

After 4 weeks of intervention it was observed a 28.4% (39/132) (CI 95%: 20.9–37.0) reduction of leg cramps in all participants and no difference between the two groups was found;

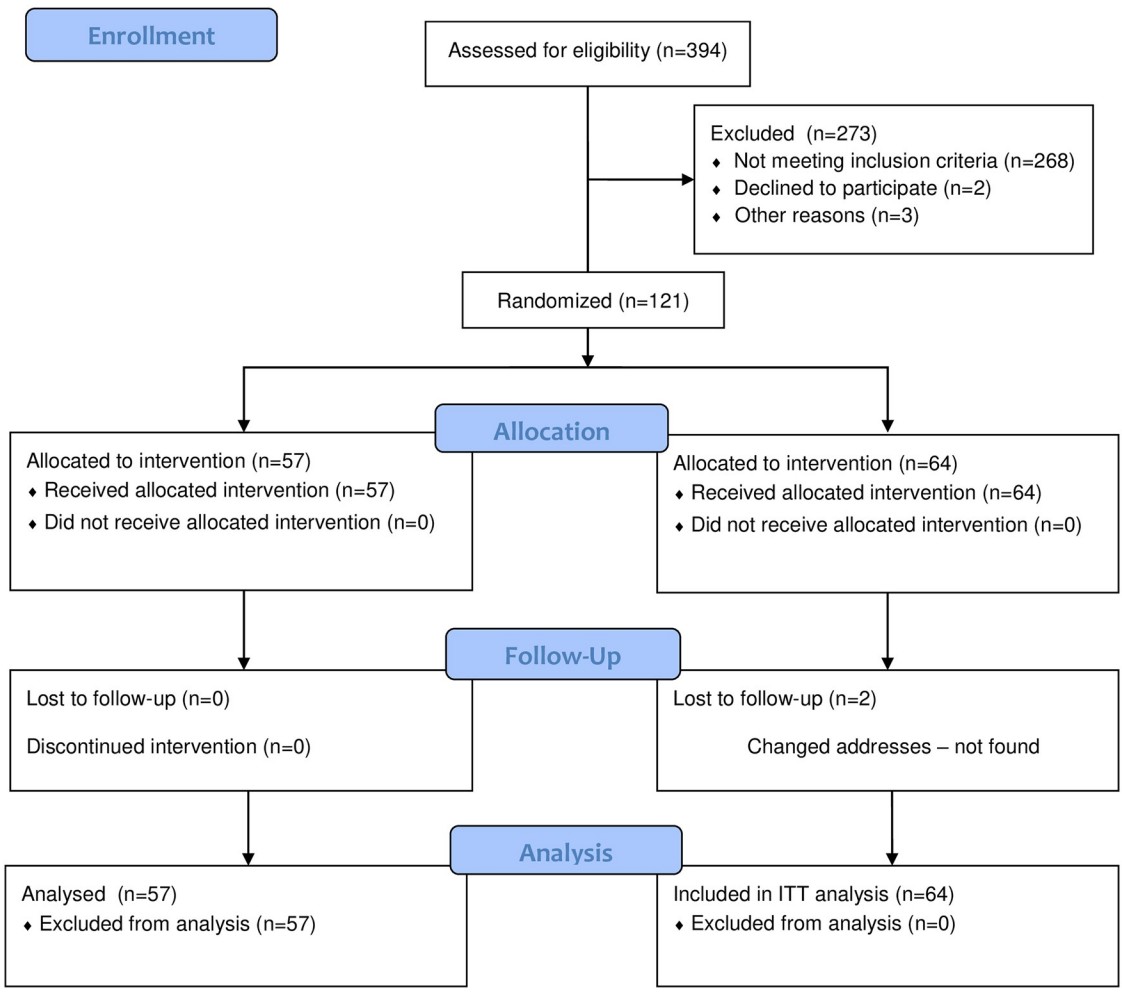

**Fig 1. CONSORT flow diagram.**

**Table 1. Some sociodemographic and clinical characteristics of the participants.**

| | Participants (132) | Mg++ group (66) | Placebo group (66) | p-value |
|---|---|---|---|---|
| **Age (years)** | 26.6± 5.5 | 26.2± 4.9 | 27.0.± 6.0 | 0.407 |
| **Years of study** | 4.8± 1.1 | 5.0± 1.2 | 4.7± 1.0 | 0.204 |
| **Currently employed** | 68 (51.5%) | 36 (54.5%) | 32 (48.4%) | 0.486 |
| **Income *per capita monthly* (US$)** | 144 ± 65 | 153 ±40 | 135 ±65 | 0.107 |
| *Primipara* | 51 (38.6%) | 30 (45.4%) | 21 (31.8%) | 0.568 |
| **Body Mass Index (BMI)** | 25.9±5.2 | 26.5± 5.8 | 25.2±4.4 | 0.173 |
| **Gestational age (weeks)** | 15.0±3.5 | 15.0± 3.2 | 15.1± 3.9 | 0.576 |
| **Number of cramps per week** | 4.8±2.8 | 4.1±2.9 | 5.5±3.0 | 0.300 |
| **Serum magnesium level < 1.8 mg/dl** | 66 (50.0%) | 32 (48.4%) | 34 (51.5%) | 0.431 |

reduction of 27.2% (18/66) (CI 95%: 17.0–39.6) in Mg++ group and a reduction of 32.8% (21/66) (CI 95%: 21.6–45.7) in the placebo group. The OR of leg cramps was 1.3 (CI 95%: 0.6–2.9), p = 0.527, taking the placebo group as reference. Among pregnant women who remained with leg cramps the number of leg cramps episodes per week showed no significance difference between the Mg++ and placebo groups (t-Student test: p = 0.408) (Table 2). The mean magnesium serum level was 1.84 mg/dL (CI 95%: 1.80–1.87) in the Mg++ group and 1.84 mg/dL (CI 95%: CI 1.80–1.87) in placebo group (t-Student test: p = 0.872).

Four pregnant women showed gastrointestinal side effects; 2 in each group had nauseas and diarrhoea.

## Discussion

This observational controlled trial showed no efficacy of oral magnesium supplementation in the treatment of leg cramps during pregnancy. Oral magnesium treatment for leg cramps during pregnancy is still a controversial issue and only a few randomized controlled trials have accessed this intervention. Recently a Cochrane systematic review concluded that magnesium supplements did not consistently reduce how often women experienced leg cramps when compared with placebo [2].

Our results were similar with Nygaard et al [14]. This randomized controlled trial assessed the effect of two weeks of oral magnesium (360 mg) on leg cramps in 38 pregnant women. Leg cramp frequency and intensity were not influenced by oral magnesium supplementation.

However other studies showed different results. Dahle et al studied 73 women with pregnancy-related leg cramps in a randomized trial and oral magnesium for 3 weeks decreased leg cramp distress [15]. Supakatisant & Phupong in a RCT studied 80 pregnant women for 4 weeks [16] and observed a fifty per cent reduction of cramp frequency and intensity in the oral magnesium bisglycinate chelate group (300mg/day). Zarean & Tarjan verified that pregnant women with low magnesium level supllemented with magnesium (200 mg) had less leg cramps during pregnancy [17].

**Table 2. Leg cramps after 4 weeks of intervention.**

| | Mg++ (66) | Placebo (66) | P |
|---|---|---|---|
| Leg cramps | | | |
| *Yes* | 48 (73.8%) | 45 (68.2%) | 0.352 |
| *No* | 18 (27.2%) | 21 (31.8%) | |
| Number of episodes per week (95% CI) | 4.1 (2.4–7.1) | 4.8 (3.7–10.0) | 0.300 |

These trials have different measurements. Pregnant women with different gestational ages were studied, interventions for variable periods of time (2 to 4 weeks) were used, different oral magnesium doses and oral formulations were administered. In addition, the definition of leg cramps was not well stablished in some studies. All this may explain different findings.

Gastrointestinal side effects were observed in a few women and there were no differences between the magnesium and placebo groups. The Cochrane systematic review concluded that there was no difference in the experience of side effects, such as nausea and diarrhoea [2].

The pregnant women in this study had a magnesium serum level in the minimum limit of normality (1.8 mg/dl) and around half of participants had hypomagnesemia. This seems to indicate that the pregnant women studied were at nutritional risk. The 300 mg daily dose of magnesium citrate used herein approximated that recommended in pregnancy [7]. Although magnesium deficiency has been implicated with an increased risk for gestational and adverse perinatal outcomes, there is not enough high-quality evidence to show that dietary magnesium supplementation during pregnancy is beneficial. [2, 7, 9].

Our study has strengths and limitation. Our strengths include: 1) A large number of participants and a very low rate of dropout; 2) Magnesium serum level was assessed; 3) Intention-to-treat analysis.

The current sample size might not be sufficient for detecting the differences in the primary outcome, since they calculated it according to the frequency of leg cramp. Additionally, the 50% reduction in frequency of leg cramp in treatment group might be too positive. All these above might be the reasons why this study had a negative result. Other limitations are as follows: 1) The intensity of leg cramps pain was not evaluated. However this evaluation includes a high degreee of subjectivity; 2) Magnesium serum level was not assessed after intervention; 3) Pregnancy women were studied in the first trimester of pregnancy but leg cramps is a more common problem in the third trimester; 4) Because this study was observational, it could be prone to biases.

In conclusion, oral magnesium supplementation during pregnancy did not reduce the ocurrence and frequency of episodes of leg cramps.

## Supporting information

**S1 Fig. Original clinical protocol in Portuguese.**
(TIF)

**S2 Fig. CONSORT 2010 checklist.**
(TIF)

**S3 Fig. Protocol English.**
(TIF)

**S1 Dataset.**
(DTA)

## Author Contributions

**Conceptualization:** Carla Adriane Leal de Araújo, João Guilherme B. Alves.

**Data curation:** Carla Adriane Leal de Araújo, Suélem Barros de Lorena, Guilherme Camelo de Sousa Cavalcanti, Gabriel Landim de Souza Leão, Geraldo Padilha Tenório.

**Formal analysis:** Carla Adriane Leal de Araújo, Suélem Barros de Lorena, Guilherme Camelo de Sousa Cavalcanti, Gabriel Landim de Souza Leão, Geraldo Padilha Tenório, João Guilherme B. Alves.

**Funding acquisition:** João Guilherme B. Alves.

**Investigation:** Carla Adriane Leal de Araújo, Suélem Barros de Lorena, Guilherme Camelo de Sousa Cavalcanti, Gabriel Landim de Souza Leão, Geraldo Padilha Tenório, João Guilherme B. Alves.

**Methodology:** Carla Adriane Leal de Araújo, Guilherme Camelo de Sousa Cavalcanti, Gabriel Landim de Souza Leão, Geraldo Padilha Tenório, João Guilherme B. Alves.

**Project administration:** Carla Adriane Leal de Araújo.

**Resources:** João Guilherme B. Alves.

**Supervision:** Carla Adriane Leal de Araújo, João Guilherme B. Alves.

**Writing – original draft:** Carla Adriane Leal de Araújo, Suélem Barros de Lorena, João Guilherme B. Alves.

**Writing – review & editing:** Carla Adriane Leal de Araújo, Suélem Barros de Lorena, Guilherme Camelo de Sousa Cavalcanti, Gabriel Landim de Souza Leão, Geraldo Padilha Tenório, João Guilherme B. Alves.

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
