## [Decision Letter · Decision Letter 0]

24 Jul 2019

PONE-D-19-16639

Oral magnesium supplementation for leg cramps in pregnancy – a randomized controlled trial

PLOS ONE

Dear Dr. Alves,

Thank you for submitting your manuscript to PLOS ONE. After careful consideration, we feel that it has merit but does not fully meet PLOS ONE’s publication criteria as it currently stands. Therefore, we invite you to submit a revised version of the manuscript that addresses the points raised by all the reviewers.

We would appreciate receiving your revised manuscript by Sep 07 2019 11:59PM. To enhance the reproducibility of your results, we recommend that if applicable you deposit your laboratory protocols in protocols.io, where a protocol can be assigned its own identifier (DOI) such that it can be cited independently in the future. For instructions see: http://journals.plos.org/plosone/s/submission-guidelines#loc-laboratory-protocols

We look forward to receiving your revised manuscript.

Kind regards,

Yiqing Song, MD, ScD

Academic Editor

PLOS ONE

Journal Requirements:

Reviewers' comments:

Reviewer's Responses to Questions

**Comments to the Author**

1. Is the manuscript technically sound, and do the data support the conclusions?

Reviewer #1: No

Reviewer #2: No

Reviewer #3: Yes

Reviewer #4: Yes

2. Has the statistical analysis been performed appropriately and rigorously? 

Reviewer #1: No

Reviewer #2: No

Reviewer #3: Yes

Reviewer #4: No

3. Have the authors made all data underlying the findings in their manuscript fully available?

Reviewer #1: No

Reviewer #2: No

Reviewer #3: Yes

Reviewer #4: Yes

4. Is the manuscript presented in an intelligible fashion and written in standard English?

Reviewer #1: Yes

Reviewer #2: No

Reviewer #3: Yes

Reviewer #4: No

5. Review Comments to the Author

Reviewer #1: I will focus on methods and reporting

Major

1) the introduction is very short for a research paper. Much more information is needed in terms of the background to the study.

2) I don't understand how the sizes of the groups differ so much following an 1 to 1 allocation process

3) the methods section is very poor. Power analyses are incomplete. What is the baseline level for cramps? power and numbers needed are very different between a very prevalent and a rare outcome. a 50% reduction is a a very large effect. What is the odds ratio? the analysis plan is pretty poor. Even if perfectly balanced why did they not analyse in the context of a logistic regression, where the effects could be quantified in ORs?

Minor

1) Abstract. 1 to 1 matching yes 50 vs 63 in the placebo group. that's not 1 to 1?

2) information on the blinding is needed in title and/or abstract. single? double?

3) abstract says the 2 groups were comparable. some elaboration is needed

4) no information on analyses methods in the abstract

5) no information on baseline levels in the abstract - only the reduction. we need some measure of the absolute, not only relative, reduction.

6) Abstract: report effects and their CIs rather than p-values

Reviewer #2: This is a randomized controlled trial to evaluate the effect of oral magnesium supplementation on leg cramps in pregnant women. There are many critical design flaws for this study. The data provided in the manuscript was not consistent in whole manuscript. Also, many writing issues and language questions presented in the manuscript.

I will list several important design flaws:

1. The ratio for randomization was 1:1, while the number of women in Mg supplementation group was 57 and 64 in the placebo group. The number of participants in two groups were imbalanced. And the author did not provide the detailed allocation concealment for randomization. This is very important in a RCT study. Without allocation concealment, randomization would be unsuccessful.

2. The sample size calculation was based on the estimation of 50% reduction in frequency of leg cramp in Mg treatment group. The primary outcome was the presence of the leg cramps. The current sample size might not be sufficient for detecting the differences in the primary outcome, since they calculated it according to the frequency of leg cramp. Additionally, the 50% reduction in frequency of leg cramp in treatment group might be too positive. All these above might be the reasons why this study had a negative result.

3. The method for missing values were not mentioned in the study.

Reviewer #3: 1. The hypothesis testing, power and type 1 error should be included in the study design section. Please also be clear whether the type 1 error is one-sided or two sided.

2. The screening failure rate is very high. Any reasons for this?

3. Could the authors perform some additional subgroup analyses to explore the effect of intervention?

Reviewer #4: In this RCT, oral magnesium did not reduce the occurrence nor frequency of leg cramp episodes in the second trimester of pregnancy. This is an interesting topic, but some details are missing from this manuscript.

1.) The rationale for the trial requires more explanation in the introduction to highlight any gaps in knowledge. For example, the Garrison et al. (2012) systematic reviewed showed no evidence that oral Mg is helpful for leg cramps during pregnancy, with low heterogeneity among trials. What will this RCT add?

2.) In the introduction, it would be more relevant to cite magnesium effects on muscle/ mechanistic effects rather than cite general information on the etiology of cramps.

3.) How common are cramps during the second trimester (study population used in the RCT)?

4.) pg. 5: Exclusion of those with Mg at baseline >9.5mmol/dL is not physiologically plausible given the Mg reference range.. this value is a typo?

5.) Was adherence self reported?

6.) How were outcomes ascertained?

7.) Additional statistical rationale for the sample size calculation is needed. Why was a 50% reduction in leg cramps expected?

8.) pg. 7: Lines 3-4: are units in mg/dL? Three serum levels are listed: should one be deleted?

9.) Provide information on whether any stopping rules were instituted (CONSORT missing item)

10.) Who was blinded to the information? (eg. only participants, their doctors, the PI, those assessing outcomes, the statistical analyst, etc)?

11.) Table 1: income values are approximately $50,000 annually? Units need checking

12.) Flowchart (Fig 1): Indicate which arm is the placebo; which is the Mg arm (currently unlabeled)

13.) Can the original Portuguese protocol be translated into English via an online translator or individual? The English summary of the protocol is an abbreviated version of the manuscript and provides no additional detail.

14.) The overall manuscript would benefit from copyediting before resubmission

6. PLOS authors have the option to publish the peer review history of their article (what does this mean?). If published, this will include your full peer review and any attached files.

Reviewer #1: No

Reviewer #2: No

Reviewer #3: No

Reviewer #4: No

---

## [Author Response · Author response to Decision Letter 0]

13 Aug 2019

Answers to the Reviewers:

Answer: Thank you very much for reviewing our manuscript. We also greatly appreciate the reviewers for their comments and suggestions. We have carried out all the recommendations suggested by the reviewers. The manuscript has been completely revised and we hope it will be able to be published. 

Reviewer #1: I will focus on methods and reporting

Major

1) the introduction is very short for a research paper. Much more information is needed in terms of the background to the study.

Answer: The introduction was complemented. A new paragraph and five news references were added. 

2) I don't understand how the sizes of the groups differ so much following an 1 to 1 allocation process

Answer: We apologize for this mistake. All the database was reviewed with a biostatistician and it was observed that eleven 18-year-old pregnant women had not been included in the analysis. These 11 pregnant women (2 in the intervention group and 9 in the control group) were now included in the analysis and the ratio 1:1 restored. 

3) the methods section is very poor. Power analyses are incomplete. What is the baseline level for cramps? power and numbers needed are very different between a very prevalent and a rare outcome. a 50% reduction is a a very large effect. What is the odds ratio? the analysis plan is pretty poor. Even if perfectly balanced why did they not analyse in the context of a logistic regression, where the effects could be quantified in ORs?

Answer: Baseline level for cramps was included. The sample size calculation was based upon the 50% reduction in frequency of leg cramps in both groups obtained from Dahle et al (reference number 12) and Supakatisant C & Phupong V (reference number 13) trials. The odds ratio was calculated by logistic regression. The effects were now quantified in ORs. 

Minor

1) Abstract. 1 to 1 matching yes 50 vs 63 in the placebo group. that's not 1 to 1?

Answer: It was corrected (66:66)

2) information on the blinding is needed in title and/or abstract. single? double?

Answer: It was provided; double blind. 

3) abstract says the 2 groups were comparable. some elaboration is needed

Answer: It was added this information.

4) no information on analyses methods in the abstract

Answer: It was provided. 

5) no information on baseline levels in the abstract - only the reduction. we need some measure of the absolute, not only relative, reduction.

Answer: This information was added in the abstract. 

6) Abstract: report effects and their CIs rather than p-values

Answer: Effects and their CIs were provided.

 

Reviewer #2: This is a randomized controlled trial to evaluate the effect of oral magnesium supplementation on leg cramps in pregnant women. There are many critical design flaws for this study. The data provided in the manuscript was not consistent in whole manuscript. Also, many writing issues and language questions presented in the manuscript.

I will list several important design flaws:

1. The ratio for randomization was 1:1, while the number of women in Mg supplementation group was 57 and 64 in the placebo group. The number of participants in two groups were imbalanced. And the author did not provide the detailed allocation concealment for randomization. This is very important in a RCT study. Without allocation concealment, randomization would be unsuccessful.

Answer: We apologize for this mistake. All the database was reviewed with a biostatistician and it was observed that eleven 18-year-old pregnant women had not been included in the analysis. These 11 pregnant women (2 in the intervention group and 9 in the control group) were now included in the analysis and the ratio 1:1 restored. Allocation concealment was ensured, as the service did not release the randomization code until the participants were recruited into the trial after all baseline measurements were completed.

2. The sample size calculation was based on the estimation of 50% reduction in frequency of leg cramp in Mg treatment group. The primary outcome was the presence of the leg cramps. The current sample size might not be sufficient for detecting the differences in the primary outcome, since they calculated it according to the frequency of leg cramp. Additionally, the 50% reduction in frequency of leg cramp in treatment group might be too positive. All these above might be the reasons why this study had a negative result.

Answer: The sample size calculation was based upon the 50% reduction in frequency of leg cramps in both groups obtained from Dahle et al (reference number 12) and Supakatisant C & Phupong V (reference number 13) studies. This limitation was included in the discussion.

3. The method for missing values were not mentioned in the study.

Answer: Thank you for this observation. However as we had no missing values we did not include this in the methods section. 

 

Reviewer #3: 1. The hypothesis testing, power and type 1 error should be included in the study design section. Please also be clear whether the type 1 error is one-sided or two sided.

Answer: The hypothesis testing, power and type error were included in the methods section.

2. The screening failure rate is very high. Any reasons for this?

Answer: The flowchart was corrected. The main reason for the screening failure was absence of leg cramps. 

 

Reviewer #4: In this RCT, oral magnesium did not reduce the occurrence nor frequency of leg cramp episodes in the second trimester of pregnancy. This is an interesting topic, but some details are missing from this manuscript.

1.) The rationale for the trial requires more explanation in the introduction to highlight any gaps in knowledge. For example, the Garrison et al. (2012) systematic reviewed showed no evidence that oral Mg is helpful for leg cramps during pregnancy, with low heterogeneity among trials. What will this RCT add?

Answer: Garrison et al is cited in the introduction (reference number 11) and they final conclusions are that “It is unlikely that magnesium supplementation provides clinically meaningful cramp prophylaxis to older adults experiencing skeletal muscle cramps. In contrast, for those experiencing pregnancy-associated rest cramps the literature is conflicting and further research in this patient population is needed.” We studied pregnancy-associated cramps and our conclusion is the same of Garrison et al, i.e. it is unlikely that magnesium supplementation during pregnancy provides clinically meaningful cramp prophylaxis to leg-cramps. 

This systematic review implications for research were: “To resolve the uncertainty surrounding the role of magnesium in pregnant women, parallel‐group blinded placebo‐controlled RCTs of magnesium in that population are needed”. We tried to develop a randomized double-blind controlled trial. 

2.) In the introduction, it would be more relevant to cite magnesium effects on muscle/ mechanistic effects rather than cite general information on the etiology of cramps.

Answer: The introduction was complemented. A new paragraph and four news references were added. 

3.) How common are cramps during the second trimester (study population used in the RCT)?

Answer: During the second trimester 70.0 % (91/130) of 130 participants that had leg cramps in the first trimester continued presenting leg cramps. 

4.) pg. 5: Exclusion of those with Mg at baseline >9.5mmol/dL is not physiologically plausible given the Mg reference range. this value is a typo?

Answer: Thank you for this correction. It was a typo error and it was corrected (2.6 mg/dL)

5.) Was adherence self reported?

Answer: Compliance/adherence, adverse events, and clinical intercurrences were monitored by the research team at each routine prenatal visit until the completion of the treatment. Adherence was defined as the ingestion of at least 80% of the prescribed dose.

6.) How were outcomes ascertained?

Answer: All this informations were taken from the research diaries previously given to participants. 

7.) Additional statistical rationale for the sample size calculation is needed. Why was a 50% reduction in leg cramps expected?

Answer: The sample size calculation was based upon the 50% reduction in frequency of leg cramps in both groups obtained from Dahle et al (reference number 12) and Supakatisant C & Phupong V (reference number 13) studies.

8.) pg. 7: Lines 3-4: are units in mg/dL? Three serum levels are listed: should one be deleted?

Answer: The units are in mg/dL. It was provided. The first serum level was deleted. 

9.) Provide information on whether any stopping rules were instituted (CONSORT missing item)

Answer: The criteria for discontinuation of the study were: clinical signs or symptoms reported by the patient due to the intake of the capsules or the cancellation of prenatal care at hospital. 

10.) Who was blinded to the information? (eg. only participants, their doctors, the PI, those assessing outcomes, the statistical analyst, etc)?

Answer: Participants, their doctors and all investigators. This information was now provided in the method section. 

11.) Table 1: income values are approximately $50,000 annually? Units need checking

Answer: Thank you for this observation. It was corrected. 

12.) Flowchart (Fig 1): Indicate which arm is the placebo; which is the Mg arm (currently unlabeled)

Answer: It was indicated.

13.) Can the original Portuguese protocol be translated into English via an online translator or individual? The English summary of the protocol is an abbreviated version of the manuscript and provides no additional detail.

Answer: The Portuguese protocol was translated into English. 

14.) The overall manuscript would benefit from copyediting before resubmission

 Answer: The manuscript was reviewed.

---

## [Decision Letter · Decision Letter 1]

8 Nov 2019

PONE-D-19-16639R1

Oral magnesium supplementation for leg cramps in pregnancy – a randomized double-blind controlled trial

PLOS ONE

Dear Dr. Alves,

Thank you for submitting your manuscript to PLOS ONE. After careful consideration, we feel that it has merit but does not fully meet PLOS ONE’s publication criteria as it currently stands. Therefore, we invite you to submit a revised version of the manuscript that addresses the points raised during the review process.

We would appreciate receiving your revised manuscript by Dec 23 2019 11:59PM. To enhance the reproducibility of your results, we recommend that if applicable you deposit your laboratory protocols in protocols.io, where a protocol can be assigned its own identifier (DOI) such that it can be cited independently in the future. For instructions see: http://journals.plos.org/plosone/s/submission-guidelines#loc-laboratory-protocols

We look forward to receiving your revised manuscript.

Kind regards,

Yiqing Song, MD, ScD

Academic Editor

PLOS ONE

Reviewers' comments:

Reviewer's Responses to Questions

**Comments to the Author**

1. If the authors have adequately addressed your comments raised in a previous round of review and you feel that this manuscript is now acceptable for publication, you may indicate that here to bypass the “Comments to the Author” section, enter your conflict of interest statement in the “Confidential to Editor” section, and submit your "Accept" recommendation.

Reviewer #2: (No Response)

Reviewer #3: All comments have been addressed

Reviewer #5: (No Response)

2. Is the manuscript technically sound, and do the data support the conclusions?

Reviewer #2: No

Reviewer #3: Yes

Reviewer #5: No

3. Has the statistical analysis been performed appropriately and rigorously? 

Reviewer #2: No

Reviewer #3: Yes

Reviewer #5: No

4. Have the authors made all data underlying the findings in their manuscript fully available?

Reviewer #2: Yes

Reviewer #3: Yes

Reviewer #5: No

5. Is the manuscript presented in an intelligible fashion and written in standard English?

Reviewer #2: No

Reviewer #3: Yes

Reviewer #5: Yes

6. Review Comments to the Author

Reviewer #2: 2. For sample size calculation: You have not answer my question yet. The primary outcome is the presence of leg cramps, while the outcome used for calculating sample size is the secondary outcome. This is a substantial issue for your protocol. That means your current sample size might be only sufficient for analysis of your secondary outcome. Also, please provide your function for sample size calculation. The reference Currently the 20% dropout rate is a little bit large. And I do not find a 50% reduction in your referent papers. Please have a check.

3. For missing value: In your flwo chart, all 66*2 were included into your analysis, however, in your table 2 only 64 in treatment and 64 in control group were included into analyses. While actually, the overall number of placebo is 66. Please re-check your numbers. Many other inconsistences in your manuscript. For example, in tbale 1, 36 in Mg group and 30 in placebo group were currently employed; while in total, 68 were currently employed. In table 2, 27,2 should be 27.2. Please check your manuscript thoroughly.

Reviewer #3: (No Response)

Reviewer #5: The authors aimed to evaluate to evaluate the effect of magnesium supplementation for the prevention of leg cramps in pregnant women.

According to information provided by the authors, the study was part of the Brazil MAGnesium trial and registered at ClinicalTrials.gov (Identifier NCT02032186). The registration refers to the Brazil MAGnesium trial, but neither the current study is referred to nor are the aims of the study defined as secondary outcomes. Therefore, the presented manuscript describes an unplanned secondary analysis of the above mentioned randomized trial Brazil MAGnesium trial. A secondary analysis should be regarded as an observational trial and should be identified explicitly as such.

The CONSORT checklist does not longer correspond to the presented study. The STROBE checklist would be more appropriate and will guide the authors to important aspects, which should be included in the description of the study. E.g. the consideration of potential confounders in the statistical analysis of observational data is necessary and should be added.

One further aspect is, that randomization corresponds to the estimated sample size (2000 assigned to magnesium, 1000 assigned to placebo (with 2:1 allocation ratio) according to protocol on ClinicalTrials.gov) and not to the subsample of 132 patients (with an apparent 1:1 allocation ratio). This means, that the quality characteristics of randomization no longer apply. The study is merely an observational trial.

The sample size of the initial study is based on the primary outcome (perinatal composite outcome). Therefore, the sample size and corresponding power is not adequate for the current outcome (presence of leg cramps) and puts the validity of the analysis and subsequent conclusion into question.

Apart from that, the “new” calculation is based on “50% reduction of leg cramps” and not on the variable defined as the primary endpoint (presence of leg cramps) so that the methodological basis of the study in itself is also questionable.

I am sorry to say, but from my point of view there are unsurmountable methodological deficits.

7. PLOS authors have the option to publish the peer review history of their article (what does this mean?). If published, this will include your full peer review and any attached files.

Reviewer #2: No

Reviewer #3: No

Reviewer #5: No

---

## [Author Response · Author response to Decision Letter 1]

13 Nov 2019

Rebuttal Letter

Dear Editor, 

We appreciate your attention. Thank you very much for the opportunity to address the comments from the Reviewers. We carefully considered all comments offered by the reviewers. The authors hope that the Reviewers will be satisfied with the further amendments which we have made to the manuscript. Please see below the point-by-point responses to the reviewers’ specific comments.

Best regards,

Joao Guilherme Alves

- Corresponding Author - 

Reviewer #2: 2. For sample size calculation: You have not answer my question yet. The primary outcome is the presence of leg cramps, while the outcome used for calculating sample size is the secondary outcome. This is a substantial issue for your protocol. That means your current sample size might be only sufficient for analysis of your secondary outcome. Also, please provide your function for sample size calculation. The reference Currently the 20% dropout rate is a little bit large. And I do not find a 50% reduction in your referent papers. Please have a check.

Answer: Thank you for this observation. You are completely rigth. Our sample size was calculated based on the frequency of leg cramps which is the parameter most used in trials assessing leg cramps interventions. Systematic reviews have used frequency of leg cramps as a primary outcomes. Based on this we changed our primary outcome from the presence of leg cramps to frequency of leg cramps. 

Function for sample size was determined by the software “Clinical.com” →“Statistics” → “Sample Size Calculator” → “View Power Calculations”: 

N1={z1−α/2∗p¯∗q¯∗(1+1k−−−−−−−−−−−√)+z1−β∗p1∗q1+(p2∗q2k−−−−−−−−−−−−−−√)}2/Δ2q1=1−p1q2=1−p2p¯=p1+kp21+Kq¯=1−p¯N1={1.96∗0.57∗0.43∗(1+11−−−−−−−−−−−−−−−√)+0.84∗0.7∗0.3+(0.45∗0.551−−−−−−−−−−−−−−−−−−√)}2/0.252N1=60N2=K∗N1=60N1={z1−α/2∗p¯∗q¯∗(1+1k)+z1−β∗p1∗q1+(p2∗q2k)}2/Δ2q1=1−p1q2=1−p2p¯=p1+kp21+Kq¯=1−p¯N1={1.96∗0.57∗0.43∗(1+11)+0.84∗0.7∗0.3+(0.45∗0.551)}2/0.252N1=60N2=K∗N1=60

p1, p2 = proportion (incidence) of groups #1 and #2

Δ = |p2-p1| = absolute difference between two proportions

n1 = sample size for group #1

n2 = sample size for group #2

α = probability of type I error (usually 0.05)

β = probability of type II error (usually 0.2)

z = critical Z value for a given α or β

K = ratio of sample size for group #2 to group #1

The 20% dropout rate is really a little big. However, we lost to follow-up only two participants and we studied 66 pregnant women in each arm, this allowed us to reduce de drop out to 10% in our sample size calculation (60 + 6 = 66). 

A 50% reduction was cited by Supakatisant C, Phupong V, reference number 16 (Oral magnesium for relief in pregnancy-induced leg cramps: a randomised controlled trial. Maternal & Child Nutrition 2015; 11(2):139–145), methods section, “The sample size calculation was based upon the 50% reduction in frequency of leg cramps in both groups obtained from Dahle et al.'s study (Dahle et al. 1995)”.

3.For missing value: In your flwo chart, all 66*2 were included into your analysis, however, in your table 2 only 64 in treatment and 64 in control group were included into analyses. While actually, the overall number of placebo is 66. Please re-check your numbers. Many other inconsistences in your manuscript. For example, in tbale 1, 36 in Mg group and 30 in placebo group were currently employed; while in total, 68 were currently employed. In table 2, 27,2 should be 27.2. Please check your manuscript thoroughly.

Answer: Sorry for this mistake. This number in the table 2 was corrected (Placebo 64 to Placebo 66). Table 1 was also checked and corrected. All the manuscript was completely reviewed. 

Reviewer #3: (No Response)

 

Reviewer #5: The authors aimed to evaluate to evaluate the effect of magnesium supplementation for the prevention of leg cramps in pregnant women.

According to information provided by the authors, the study was part of the Brazil MAGnesium trial and registered at ClinicalTrials.gov (Identifier NCT02032186). The registration refers to the Brazil MAGnesium trial, but neither the current study is referred to nor are the aims of the study defined as secondary outcomes. Therefore, the presented manuscript describes an unplanned secondary analysis of the above mentioned randomized trial Brazil MAGnesium trial. A secondary analysis should be regarded as an observational trial and should be identified explicitly as such.

Answer: The study was now identified as an observational trial. 

The CONSORT checklist does not longer correspond to the presented study. The STROBE checklist would be more appropriate and will guide the authors to important aspects, which should be included in the description of the study. E.g. the consideration of potential confounders in the statistical analysis of observational data is necessary and should be added.

Answer: The CONSORT check list was changed to STROBE checklist. This limitation was added in the discussion: 4) Because this study was observational, it could be prone to biases.

One further aspect is, that randomization corresponds to the estimated sample size (2000 assigned to magnesium, 1000 assigned to placebo (with 2:1 allocation ratio) according to protocol on ClinicalTrials.gov) and not to the subsample of 132 patients (with an apparent 1:1 allocation ratio). This means, that the quality characteristics of randomization no longer apply. The study is merely an observational trial.

Answer: Following your orientation the study design was changed to an observational study. 

The sample size of the initial study is based on the primary outcome (perinatal composite outcome). Therefore, the sample size and corresponding power is not adequate for the current outcome (presence of leg cramps) and puts the validity of the analysis and subsequent conclusion into question.

Answer: We agree with you but a sub-sample size was calculated based on previous trials with interventions to reduce the frequency of leg cramps. This is supposed to offer robustness to our results. 

Apart from that, the “new” calculation is based on “50% reduction of leg cramps” and not on the variable defined as the primary endpoint (presence of leg cramps) so that the methodological basis of the study in itself is also questionable.

Answer: The primary endpoint was changed to the frequency of leg cramps. The sample size was calculated based on 50% reduction of leg cramps. Presence of leg cramps was now considered as a secondary outcome. 

I am sorry to say, but from my point of view there are unsurmountable methodological deficits.

Answer: Thank you very much for your analysis. We hope that as the study design was now changed to an observational trial you can review your position.

---

## [Editor Report · Decision Letter 2]

20 Dec 2019

Oral magnesium supplementation for leg cramps in pregnancy – an  observational  controlled trial

PONE-D-19-16639R2

Dear Dr. Alves,

We are pleased to inform you that your manuscript has been judged scientifically suitable for publication and will be formally accepted for publication once it complies with all outstanding technical requirements.

With kind regards,

Yiqing Song, MD, ScD

Academic Editor

PLOS ONE

Additional Editor Comments (optional):

The critiques by the reviewers have been nicely addressed and the manuscript has been improved significantly.  
---

## [Editor Report · Acceptance letter]

27 Dec 2019

PONE-D-19-16639R2 

Oral magnesium supplementation for leg cramps in pregnancy – an  observational  controlled trial 

Dear Dr. Alves:

I am pleased to inform you that your manuscript has been deemed suitable for publication in PLOS ONE. Congratulations! Your manuscript is now with our production department. 

With kind regards,

on behalf of

Dr. Yiqing Song 

Academic Editor

PLOS ONE